# Interventional Oncolytic Immunotherapy with LTX-315 for Residual Tumor after Incomplete Radiofrequency Ablation of Liver Cancer

**DOI:** 10.3390/cancers14246093

**Published:** 2022-12-11

**Authors:** Guanhui Zhou, Xuefeng Kan, Feng Zhang, Hongxiu Ji, Junhui Sun, Xiaoming Yang

**Affiliations:** 1Image-Guided Bio-Molecular Intervention Research and Section of Vascular and Interventional Radiology, Department of Radiology, University of Washington School of Medicine, 850 Republican Street, S470, Seattle, WA 98109, USA; 2Hepatobiliary and Pancreatic Interventional Treatment Center, Division of Hepatobiliary and Pancreatic Surgery, The First Affiliated Hospital, Zhejiang University School of Medicine, Hangzhou 310003, China; 3Department of Radiology, Union Hospital, Tongji Medical College, Huazhong University of Science and Technology, Wuhan 430022, China; 4Department of Pathology, Overlake Medical Center and Incyte Diagnostics, Bellevue, WA 980004, USA

**Keywords:** radiofrequency ablation, radiofrequency hyperthermia, residual tumors, liver cancer, oncolytic immunotherapy

## Abstract

**Simple Summary:**

Radiofrequency ablation (RFA) is a favorite treatment approach for patients with liver cancer, one of the most common malignancies worldwide. However, incomplete RFA often occurs in irregular and medium-to-larger (>3 cm) hepatic tumors. The aim of this study was to validate the feasibility of interventional oncolytic immunotherapy with LTX-315 for residual tumors after incomplete RFA of liver cancers. LTX-315, injected into tumor margins through the electrode prongs during the ablation procedure, can directly kill tumor cells and activate an anti-tumor immune response. This treatment strategy facilitated the creation of a clear ablated tumor margin. The evidence of this study may open up new avenues to prevent residual tumors after RFA of irregular and medium-to-large liver cancers.

**Abstract:**

Objective: To investigate the feasibility of interventional oncolytic immunotherapy with LTX-315 for residual tumors after incomplete radiofrequency ablation (iRFA) of VX2 liver tumors in a rabbit model. Methods: For in vitro experiments, VX2 tumor cells were treated with: (1) phosphate buffered saline, (2) radiofrequency hyperthermia (RFH), (3) LTX-315, and (4) RFH plus LTX-315. The residual tumors after iRFA of VX2 liver tumors were treated with: (1) phosphate buffered saline served as control, (2) 2 mg LTX-315, and (3) 4 mg LTX-315. MTS assay, fluorescence microscopy, and flow cytometry were used to compare cell viabilities and apoptosis among different groups. Ultrasound imaging was used to follow up the tumor growth, which were correlated with the optical imaging and subsequent histology. Results: For in vitro experiments, compared with the other three groups, MTS assay demonstrated the lowest cell viability, fluorescence microscopy showed the least survival cells, and apoptosis analysis revealed the highest percentage of apoptosis cells in the combination treatment groups (*p* < 0.001). For in vivo experiments, ultrasound imaging showed the smallest tumor volume in the group with 4 mg LTX-315 therapy compared with the other two groups (*p* < 0.001). The optical imaging and histopathological analysis showed complete necrosis of the tumors in the group with 4 mg LTX-315 therapy. A significant increase of CD8^+^ T cells and HSP70 and a significant decrease of Tregs were observed in residual tumors in the group with 2 mg LTX-315 therapy compared with the control group (*p* < 0.001). Conclusion: Interventional oncolytic immunotherapy with LTX-315 for residual tumors after iRFA of liver cancer is feasible, which may open up new avenues to prevent residual tumors after RFA of intermediate-to-large liver cancers.

## 1. Introduction

Liver cancer is one of the most frequent solid tumors globally [1]. As it is effective and minimally invasive, radiofrequency ablation (RFA) is a favorite treatment option for liver cancer [2]. With the current technology, RFA is most effective for small solid tumors with a diameter < 3 cm. However, for ablation of irregular and medium-to-larger tumors, RFA usually suffers from the setback of residual viable tumors at the ablated tumor margin. The reasons for this are as follows: (1) the microsatellite tumors or microvenous tumor emboli, which are located at the ablated tumor margin, cannot been detected via common imaging methods; (2) RFA heat is carried away by blood flow when the tumor is located near large blood vessels; (3) intentional avoidance of rapid overheating is necessary by the operator to protect adjacent important structures, such as the diaphragm, the gastrointestinal tract, and the gallbladder; (4) off-center positioning of the ablation electrode occurs for the limited percutaneous access windows. Moreover, previous studies [3,4,5] reported incomplete radiofrequency ablation (iRFA) could promote residual tumors metastasis and rapid progression. Attempts have been made to address this clinical issue, such as RFA in combination with transarterial chemoembolization [6]. However, the local tumor progression rate in large HCC is still up to 30.4% after such combined treatment [7]. Therefore, it is urgent to develop an effective strategy to target the residual tumors after iRFA of liver cancers and improve the long-term survival of these patients.

Oncolytic immunotherapy is a promising treatment approach for malignant tumors [8,9]. LTX-315 is a kind of oncolytic peptide and a chemically modified 9-mer cationic peptide that is derived from bovine lactoferrin [10]. It can kill tumor cells via a membranolytic effect on the cellular plasma membrane and intracellular organelles, such as mitochondria and lysosomes, which further leads to a subsequent release of danger-associated molecular pattern molecules and tumor antigens that recruit and activate T cells for immunotherapy. Due to the ability of not only indirectly killing tumor cells, but also activating an anti-tumor immune response, LTX-315 is considered a promising drug in the treatment of some malignant solid tumors, such as melanoma, breast cancers, and sarcoma [11,12].

Previous studies demonstrated radiofrequency hyperthermia (RFH) could significantly enhance the chemotherapeutic effect on a variety of malignancies by increasing the entrance of treatment agents into the targeted tumor cells [13,14,15]. Based on these, we assumed that an RFH at the ablated tumor margin may enhance the efficacy of oncolytic immunotherapy with LTX-315. Local administration of LTX-315 at the ablated tumor margin during RFA may facilitate creating a tumor-free ablated margin. To assess this hypothesis, we conducted the present study.

## 2. Materials and Methods

### 2.1. Study Design

Our study was divided into two stages: (a) in vitro experiments to validate the feasibility of using indocyanine green (ICG)-based optical imaging to assess efficacy of RFH-enhanced oncolytic immunotherapy (LTX-315) for treatment of VX2 tumor cells; (b) in-vivo experiments to validate the feasibility of intratumoral oncolytic immunotherapy (LTX-315) for residual tumors after iRFA of VX2 liver tumor in a rabbit model. All in vitro experiments were repeated six times.

### 2.2. In Vitro Evaluation

#### 2.2.1. Cell Cultures

VX2 tumor cells (IDAC, Tohoku University, Sendai, Japan) were cultured in RPMI 1640 Medium (Life Technologies Ltd., Paisley, UK) supplemented with 10% fetal bovine serum (Gibco, Grand Island, NY, USA) in an incubator at 37 °C with a 5% carbon dioxide atmosphere.

#### 2.2.2. Optimizing the Protocol for Sufficient ICG Uptake of VX2 Cells in In Vitro Experiments

VX2 tumor cells were seeded (5 × 10^4^ per well) in four-well chamber plates (Laboratory Tek II; Thermo Fisher Scientific, Rochester, NH). As cells were cultured to reach 80% confluence, VX2 tumor cells were treated with ICG at (i) concentrations of 0, 20, 40, 60, 80, 100, 120, 140, 160, 180, and 200 μg/mL for 24 h and (ii) a concentration of 100 μg/mL with various incubation times of 0, 1, 2, 4, 8, 12, 24, and 48 h. The non-treated VX2 cell groups served as control. For fluorescent microscopy, the ICG treated cells (ICG cells) were washed with phosphate buffered saline (PBS) twice to remove the free ICG, fixed with 4% paraformaldehyde, and then dried at room temperature. The cells were then counterstained with 4′,6-diamidino-2-phenylindole (DAPI; Vector Laboratories, Burlingame, CA, USA) and imaged with a fluorescent microscope (IX73, Olympus, Tokyo, Japan). For in vitro optical/X-ray imaging, the ICG cells were washed twice with PBS, trypsinized, centrifuged, and then 5 × 10^6^ cells were diluted with 0.2 mL PBS in a 96-well. Subsequently, the cell-containing wells were imaged with an optical/X-ray imaging system (In vivo Xtreme; Bruker, Billerica, MA, USA) at the emission wavelength of 830 nm and excitation wavelength of 760 nm. The fluorescent signal intensity (SI) of cells in different wells was measured and compared.

#### 2.2.3. RFH Enhanced the Killing Effect of LTX-315 for VX2 Cells

The 50% inhibitory concentration (IC50) doses of LTX-315 (27.65 μM) were used for VX2 cells treatment, which were determined via MTS assay (3-[4,5-dimethylthiazol-2-yl] II-5-[3-carboxymethoxyphenyl]-2-[4-sulfophenyl]-2H-tetrazoliu, CellTiter 96 Aqueous One Solution Cell Proliferation Assay; Promega, Madison, WI, USA). VX2 cells in different groups were treated with (1) PBS to serve as a control, (2) RFH alone at 42 °C for 30 min, (3) LTX-315 alone, and (4) RFH combined with LTX-315. In the group of RFH in combination with LTX-315, LTX-315 was added to the well of tumor cells immediately followed by 30 min RFH at 42 °C. Meanwhile, 100 μg/mL ICG was added to each group for fluorescence optical imaging. RFH was performed by attaching a custom-made 0.56 mm radiofrequency heating wire under the bottom of the chamber and connecting it to a 460 kHz radiofrequency generator (WE7568-II, Welfare Electronics Co. Beijing, China). A 400 mm fiber optical temperature probe (PhotonControl, Burnaby, BC, Canada) was placed in the chamber for temperature measurement. By adjusting the radiofrequency output power at approximately 10 W, the treatment temperature was kept at about 42 °C [13,14,15]. 

The viability of cells was evaluated via MTS assay 24 h after the treatments. The relative cell viability of different groups was calculated using the equation of A_treated_ − A_blank_/A_control_ − A_blank_, where A is absorbance.

The percentages of necrotic/apoptotic cells were quantified via flow cytometry using Annexin V-fluorescein isothiocyanate and propidium iodide (PI) staining (BD Biosciences, San Diego, CA, USA). Cells were stained with Annexin-V/FITC and PI in a binding buffer along with the appropriate control. The data were analyzed using software (FlowJo version 10.7; FloJo Data Analysis Software, Ashland, OR, USA).

#### 2.2.4. Optical Imaging of Treated Cells in Four Groups

Four groups of ICG cells in cell culture slides were washed twice with PBS (phosphate buffered saline), fixed in 4% paraformaldehyde, counterstained with 4′,6-diamidino-2-phenylindole (DAPI; Vector Laboratories, Burlingame, CA, USA), and then imaged with a fluorescence microscope. Cells in 4 groups were collected and suspended with 200 μL PBS. Then, the suspension was transferred to a 96-well plate. Bioluminescence optical imaging was performed using an optical imaging system (In-vivo Xtreme; Bruker, Billerica, MA, USA). Subsequently, the bioluminescence SI was quantified.

### 2.3. In Vivo Confirmation

#### 2.3.1. Creation of VX2 Liver Tumor in a Rabbit Model

The animal protocol was approved by the animal care and use committee of our institution. Eighteen female New Zealand rabbits (Western Oregon Rabbit Company, Philomath, OR, USA), 3–4 months old and weighing 2.0–3.0 kg, were used in this study. Each rabbit received general anesthesia with an auricular vein injection of 3% pentobarbital (30 mg/kg). Under general anesthesia, the left liver lobe of the rabbit was exposed through a subxiphoid abdominal incision, and fresh VX2 tissue (1 mm^3^) was implanted into the left lobe of the liver. The incisions were closed with layered sutures. 

#### 2.3.2. Creation of Residual Tumors after RFA and Residual Tumors Treatments

Ultrasound imaging was used to follow up the rabbits’ liver tumors growth. Two weeks after tumor implantation, VX2 liver tumors of rabbits received RFA treatment when the tumors grew to about 1 cm in diameter. To create residual tumors after RFA, the prongs of an RF electrode (Welfare Electronics Co., Beijing, China) were opened to not fully cover the tumor periphery (the tips of prong were approximately 2 mm away from the tumor periphery) under ultrasound imaging guidance, and the treatments were administered for 2.0 min with a temperature of 80 °C and output power of 30–40 W using an RFA system (We7568-II, Welfare Electronics Co., Beijing, China). Eighteen rabbits with residual VX2 tumors after iRFA treatment were randomly divided into 3 groups (*n* = 6/group). Immediately after iRFA, the prongs of the RF electrode were opened to fully cover the ablated tumor under ultrasound imaging guidance, where PBS, 2 mg LTX-315 in 100 μL PBS, or 4 mg LTX-315 in 100 μL PBS was directly infused through the prongs of the RF electrode within 3 min.

#### 2.3.3. Post-Treatment Follow-Up with Ultrasound Imaging

Ultrasound imaging was performed to assess the tumor growth at days 0, 7, and 14 after the treatment. The tumor volume was calculated using the axial, longitudinal, and depth diameters using the following equation: v= x·y·z·π/6. Data were shown as relative tumor volume (RTV) using the following equation: RTV = TV_Dn_/TV_D0_, where TV represents tumor volume, Dn represents the day after treatment, and D0 represents the day before treatment.

#### 2.3.4. Ex-Vivo Optical Imaging Confirmation

At 13 days after the treatment, all animals received intravenous administration of ICG via an auricular vein (0.5 mg/kg body weight). All the rabbits were euthanized after the last ultrasound imaging follow-up. The left liver lobe with the tumors was harvested and sliced at a thickness of 0.5–0.8 cm. The gross liver specimens were then imaged using the Bruker In-vivo Xtreme OI system at an emission wavelength of 830 nm and an excitation wavelength of 760 nm, 1.40 f-stop, field of view of 120 × 120 mm, and an exposure time of 1.0 min.

#### 2.3.5. Pathologic Correlation/Confirmation

Tumor tissues were fixed with 10% formalin, embedded in paraffin, sectioned at 4 mmol/L thickness, and analyzed with the staining of hematoxylin and eosin (H&E), terminal deoxynucleotidyl transferase-mediated 29-deoxyuridine, 59-triphosphate nick end labeling (TUNEL), Ki-67, HSP70, CD8 for CD8^+^ T cells, and Foxp-3 for Treg cells (regulatory T cells). 

### 2.4. Statistical Analysis

Statistical software (SPSS 25.0; IBM Corp, Armonk, NY, USA) was used for all data analyses. One-way analysis of variance was used to compare (a) fluorescent SI among different cell treatment groups, (b) apoptosis rate and cell viability among different cell treatment groups, and (c) relative tumor volumes and fluorescent SI among the three animal groups. The non-parametric Mann-Whitney U test was used to compare TUNEL IOD sum, KI67 IOD sum, CD8 IOD sum, HSP70 IOD sum, and Foxp3 IOD sum between the control group and the 2 mg LTX-315 therapy group. *p* values of less than 0.05 were considered statistically significant.

## 3. Results

### 3.1. In Vitro Optimization of ICG Dose and Time Window for OI of ICG Cells

Fluorescence microscopy showed that ICG cells emitted red fluorescence, which was not seen in VX2 cells without ICG treatment. With the increase of ICG concentration from 0 to 100 μg/mL, the red fluorescence signal intensity (SI) of ICG cells became increasingly stronger. Due to the quenching effect [16], the fluorescence SI decreased as the ICG concentrations increased from 100 to 200 μg/mL (Figure 1A). In vitro quantitative optical/X-ray images of ICG cells further confirmed the results of fluorescent microscopy (Figure 1B,C). Therefore, the optimum ICG concentration was determined to be 100 mg/mL to investigate the ideal time window for the detection of ICG cells in in vitro experiments.

For detection of the best time window, fluorescent microscopy showed that the red fluorescence SI become stronger as the incubation times increased from 0 to 24 to 48 h (Figure 2A). This result was confirmed via in vitro quantitative optical/X-ray images of ICG cells (Figure 2B,C). Thus, we decided that 24 h post-ICG treatment of VX2 cells was the best time window for in vitro OI of cells.

### 3.2. RFH-Enhanced Killing Effect of LTX-315 for VX2 Tumor Cells

Fluorescence microscopy performed 24 h after ICG treatments demonstrated fewer survival cells in the combination therapy group (RFH + LTX-315) compared with the other three groups (Figure 3A). The fluorescent SI was significantly decreased in the combination therapy group compared with the other three treatment groups (2.65 ± 0.18 × 10^9^ photons/sec/mm^2^ vs. 4.58 ± 0.31 × 10^9^ photons/sec/mm^2^ vs. 6.14 ± 0.42 × 10^9^ photons/sec/mm^2^ vs. 7.05 ± 0.48 × 10^9^ photons/sec/mm^2^, respectively, *p* < 0.001) (Figure 3B,D). Flow cytometry analysis further demonstrated more necrotic/apoptotic cells in the combination therapy group than the other three treatment groups (67.9 ± 5.9% vs. 44.8 ± 4.0% vs. 9.5 ± 2.4% vs. 4.1 ± 1.6%, respectively, *p* < 0.001) (Figure 3C,E). The MTS assay further quantitatively confirmed that the viability of cells in the combination group was significantly lower than that in the other three groups (25.4 ± 3.0% vs. 55.3 ± 4.4% vs. 92.7 ± 1.6% vs 97.0 ± 1.7%, *p* < 0.001) (Figure 3F). 

### 3.3. In Vivo Experiments

#### 3.3.1. Creation of Incomplete RFA Model in VX2 Liver Tumor and Residual Tumors Treatment

All rabbits with orthotopic hepatic VX2 tumors were successfully created in the left lobes of livers (Figure 4A). No significant difference in average baseline tumor volume was found among the three groups before treatment (1.03 ± 0.05 cm vs. 1.03 ± 0.08 cm vs. 0.99 ± 0.09 cm, *p* = 0.664). Under ultrasound and X-ray imaging guidance (Figure 4B–E), we successfully created residual tumors in the ablated tumor periphery. LTX-315 was directly infused through the prongs of an RF electrode to treat the residual tumors (Figure 4F).

#### 3.3.2. Ultrasound and Optical Imaging of VX2 Liver Tumors in Three Groups

All animals survived after the experimental procedures without complications. Follow-up with ultrasound images demonstrated the smallest relative tumor volume in the 4 mg LTX-315 group compared with the other two groups (0.11 ± 0.01 (4 mg LTX-315 group) vs. 0.57 ± 0.04 (2 mg LTX-315 group) vs. 6.46 ± 0.81 (control group), *p* < 0.001) (Figure 5A,B). The gross specimens of liver with the tumors were obtained at the end of the experiment. The viable residual tumor appears as white color, which was clearly outlined using ICG-based optical imaging (Figure 5A). The ex-vivo optical imaging demonstrated the lowest signal intensity of the ablated tumor in the 4 mg LTX-315 group compared with the other two groups (0.47 ± 0.06 × 10^11^ photons/sec/mm^2^ (4 mg LTX-315 group) vs. 1.78 ± 0.31× 10^11^ photons/sec/mm^2^ (2 mg LTX-315 group) vs. 8.75 ± 1.08 × 10^11^ photons/sec/mm^2^ (control group), *p* < 0.001) (Figure 5C). H&E staining further confirmed the complete necrosis of the tumor in the 4 mg LTX-315 therapy group, and there were viable residual tumors in the ablated tumor periphery in the other two groups (Figure 5A). 

### 3.4. Histology Analysis

For the complete necrosis of tumor, the histologic analysis showed there were no apoptotic cells, proliferative cells, HSP 70, CD8^+^ T cells, and Tregs in the ablated tumor areas in the 4 mg LTX-315 group. Meanwhile, the histologic analysis demonstrated there were more apoptotic cells, HSP70, and CD8^+^ T cells in the 2 mg LTX-315 group than the control group (apoptotic cells: 2.36 ± 0.12 × 10^6^ IOD sum vs. 7.99 ± 0.37 × 10^6^ IOD sum, *p* < 0.001; HSP70: 3.84 ± 0.48 × 10^6^ IOD sum vs. 10.13 ± 1.17 × 10^6^ IOD sum, *p* < 0.001; CD8^+^ T cells: 0.36 ± 0.03 × 10^6^ IOD sum vs. 3.13 ± 0.43 × 10^6^ IOD sum, *p* < 0.001) and fewer proliferating cells and Tregs in the 2 mg LTX-315 group compared with control group (proliferating cells: 3.52 ± 0.35 × 10^6^ IOD sum vs. 1.18 ± 0.22 × 10^6^ IOD sum, *p* < 0.001; Tregs: 7.60 ± 0.32 × 10^6^ IOD sum vs. 3.40 ± 0.27 × 10^6^ IOD sum) (Figure 6). 

## 4. Discussion

Incomplete RFA is attributed to the difficulties of creating a safe and effective ablation periphery with a 1 cm surgical margin (clear safety margin) beyond the tumor confinement using the current RFA technology [17,18]. The common reason for the inability to obtain this clear margin is that the thermal ablation heat is carried away by blood flow in the presence of larger neighboring vasculatures at the ablated tumor periphery (the heat-sink effect). The residual viable tumors, which are located in the non-clear ablated tumor margin after RFA treatment, often lead to tumor recurrence and persistence. In the present study, we attempted to solve this critical clinical problem by infusing LTX-315 in the ablated tumor margin through the prongs of an RF electrode. The results of our study demonstrated that infusing of sufficient LTX-315 facilitated in creating a clear ablated tumor margin. 

The heat-sink effect during RFA often results in a sublethal RFH at the ablated tumor margin. Previous studies reported that hyperthermia could increase the concentration of anti-cancer drugs in tumor cells [19,20], regulate the immune microenvironment [21], and overcome resistance mechanisms to immunotherapy in HCC [22]. Based on these reports, we delivered the oncolytic peptide of LTX-315 to the ablated tumor margin during RFA. The results of our study indicated that RFH could enhance the killing effect of LTX-315 for VX2 cells. A clear ablated tumor margin was created in the group with 4 mg LTX-315 therapy. 

LTX-315 can kill cancer cells via a membranolytic effect on the cellular plasma membrane and intracellular organelles, which further leads to a subsequent release of danger-associated molecular pattern molecules and tumor antigens that recruit and activate T cells for immunotherapy [23,24]. In our study, we found LTX-315 treatment could increase infiltration of CD8^+^ T cells and significantly decrease Tregs in the tumor bed, which altered the tumor microenvironment by turning “immune cold” tumors into “immune hot” tumors while improving the immunologically suppressive microenvironment. Meanwhile, we observed that the expression of HSP70 in residual tumors was significantly increased after LTX-315 treatment, which resulted in an effective anti-tumor effect.

As a relatively nontoxic protein-bound compound, ICG is taken up by some cells [25,26]. The detailed mechanisms for intracellular localization of ICG have not been fully clarified. Several studies indicated that liver malignant tumors can take up ICG via the organic anion transporting polypeptide 8 and transporters of Naþ/taurocholate coransporting polypeptide [27]. ICG is excreted from normal hepatocytes exclusively into the biliary system when administered intravenously. However, due to impaired biliary excretion, ICG accumulates in liver malignant tumor cells, resulting in ICG enhancement throughout the liver tumors. In the present study, we found ICG was well taken up by VX2 cells at the dose of 100 μg/mL with an incubation time of 24 h, and residual viable tumors in the ablated tumor margin displayed intense optical intensity signals after ICG-based optical imaging compared with completed ablated tumors and normal liver parenchyma, which indicated that ICG-based optical imaging may be an effective evaluation tool after RFA of liver cancer.

A primary disadvantage of systemic therapeutic administration is the insufficient dose of therapeutics delivered to the targeted tumors, resulting in limited therapeutic efficacy with dose limiting toxicity for some patients. Since LTX-315 is rapidly hydrolyzed by peptidase after entering the blood, it was not recommended for systemic therapeutic administration. In this study, we developed an innovative interventional oncology approach using image-guided local therapeutic/hyperthermia delivery to avoid limitations of systemic administration of therapeutic agents.

Our study has limitations. First, although LTX-315 was injected into tumors through the prongs of the RFA electrode, it may be unevenly distributed in the ablated tumors. Second, the VX2 tumor is derived from a papilloma virus-induced rabbit epithelial cell line. Although it can imitate human liver malignant tumor, the pathological characteristics may be different from human malignancies. Third, follow-up in our study was limited to two weeks after the treatments. Longer follow-up may result in tumor growth and extensive metastasis, which is not approved by our institutional animal care and use committee.

## 5. Conclusions

In conclusion, interventional oncolytic immunotherapy with LTX-315 for residual tumors after iRFA of liver cancer is feasible, which may open up new avenues to prevent residual tumors after RFA of intermediate-to-large liver cancers.

## Figures and Tables

**Figure 1 cancers-14-06093-f001:**
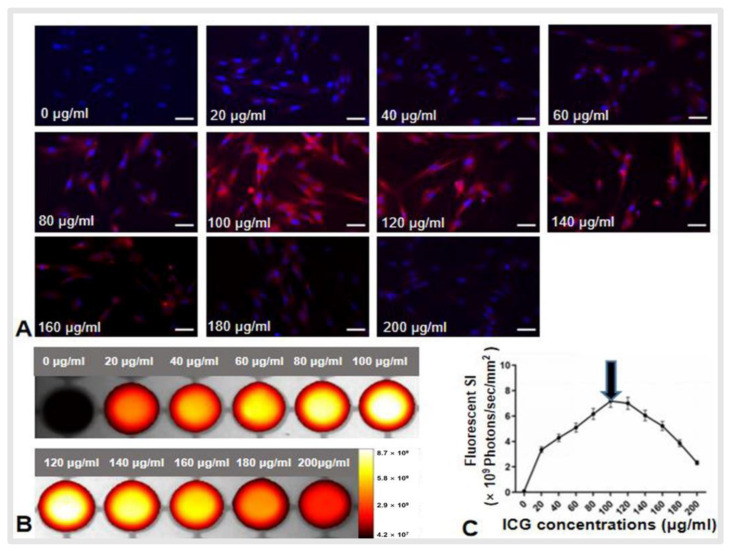
Optimization of indocyanine green (ICG) dose for optical imaging of ICG-treated VX2 cells (ICG cells). (**A**) Fluorescent microscopy images showed an increased intensity of the ICG fluorescence signals as ICG concentrations increased from 0 to 100 μg/mL and then decreased intensity as ICG concentrations increased from 100 to 200 μg/mL. Scale bars, 50 μm. (**B**) Optical/X-ray images of ICG cells showed similar results in terms of an increased fluorescent signal intensity (SI) as ICG concentrations increased from 0 to 100 μg/mL and a decreased SI as ICG concentrations increased from 100 to 200 μg/mL, (**C**) which was further confirmed via quantitative analysis of fluorescent SI.

**Figure 2 cancers-14-06093-f002:**
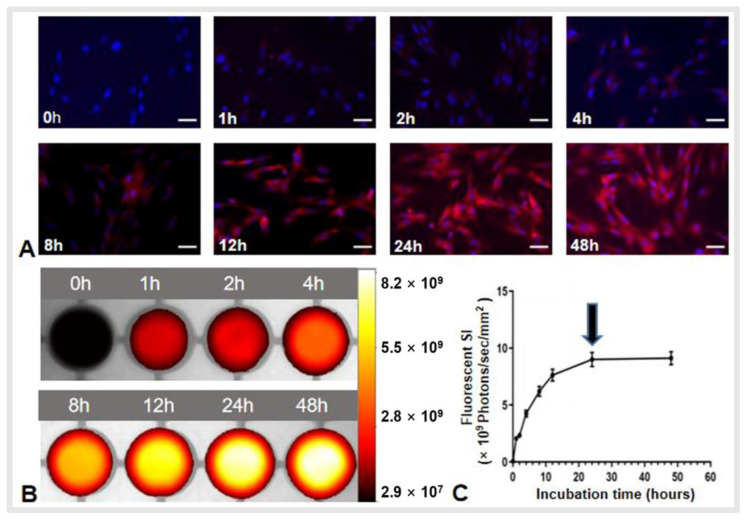
Optimization of indocyanine green (ICG) time window for optical imaging of ICG-treated VX2 tumor cells (ICG-cells). (**A**) Fluorescent microscopy showed that the red fluorescence emitted by ICG became more intense as the incubation time increased from 0 to 48 h. Scale bars, 50 μm. (**B**) Optical/X-ray images showing similar results, with the fluorescence signal of ICG cells reaching its peak at 24 h incubation time. (**C**) Quantitative analysis of the fluorescent signal intensity (SI) further confirmed the optimal time window at 24 h (arrow).

**Figure 3 cancers-14-06093-f003:**
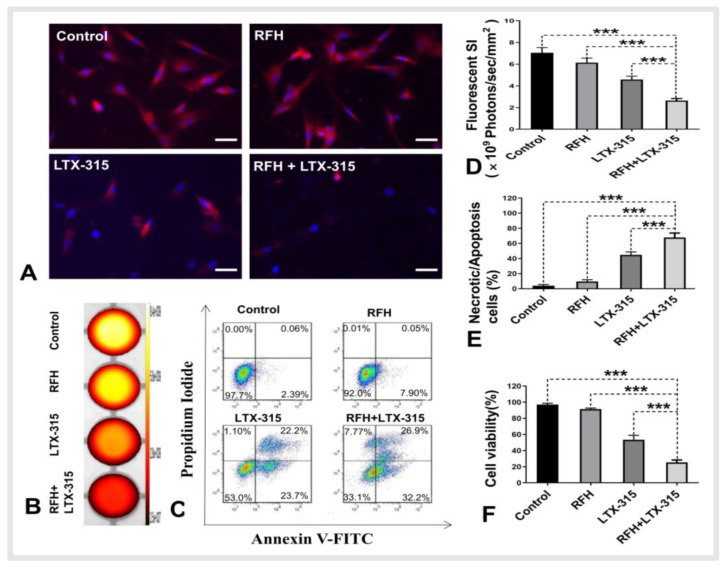
Fluorescent microscopy, optical imaging, flow cytometry, and MTS assay of the treated VX2 cells. (**A**) Fluorescent microscopy showed fewest survival VX2 cells in the combination treatment group (RFH + LTX-315), (scale bars, 50 μm), and optical imaging demonstrated the lowest fluorescent signal intensity (SI) in the combination treatment group (**B**,**D**) in comparison to the other three groups. (**C**,**E**) Flow cytometry showed the highest percentage of necrotic/apoptotic cells in the combination treatment group compared with the other three groups. (**F**) MTS assay demonstrated the lowest cell viability in the combination treatment group compared with the other three groups. (*** *p* < 0.001).

**Figure 4 cancers-14-06093-f004:**
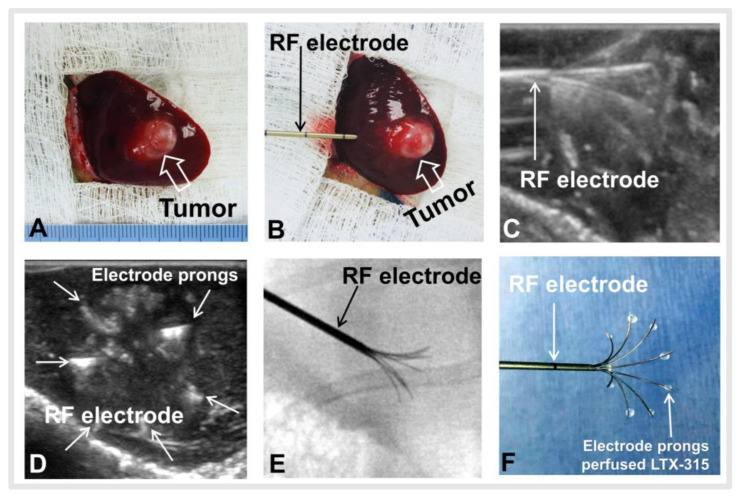
Creation of rabbit orthotopic hepatic VX2 tumor for incomplete radiofrequency ablation (RFA) and LTX-315 treatment. (**A**) An orthotopic hepatic VX2 tumor (arrow) was created in the left lobe liver of the rabbit. (**B**–**E**) Under real-time ultrasound imaging and X-ray imaging guidance, the ablation electrode was placed into the tumor and the multiple prongs were opened to not fully cover the tumor periphery to create an incomplete RFA tumor model. (**F**) The multimodal RFA electrode permitted simultaneous peritumoral infusion of LTX-315 to the residual tumors in the ablated tumor margin during RFA treatment.

**Figure 5 cancers-14-06093-f005:**
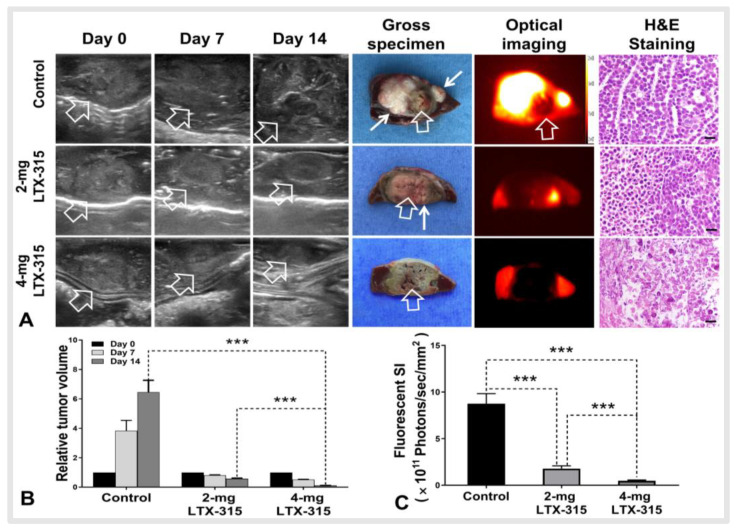
Follow-up of treated tumors among three animal groups. (**A**) The ultrasound images showed the smallest tumor size in the 4 mg LTX-315 therapy group compared with the other two groups, and the gross specimens of liver with the tumors showed the residual viable tumor in the ablated tumor margin appears as white color (small arrows), compared to RF-ablated necrotic (dead) tissues (grey-pink color, open arrows). ICG-based optical imaging detected these residual viable tumors as white-yellow color and the normal liver parenchyma surrounding the tumors as red color. H&E staining further confirmed the complete necrosis of tumor in the 4 mg LTX-315 therapy group, and there were viable residual tumor cells in the ablated tumor periphery in the other two groups (scale bars, 20 µm). (**B**) The quantitative analysis demonstrated the smallest relative tumor volume in the 4 mg LTX-315 group compared with the other two groups, and the lowest signal intensity of the ablated tumor in the 4 mg LTX-315 group, in comparison to the other two groups (**C**). (*** *p* < 0.001).

**Figure 6 cancers-14-06093-f006:**
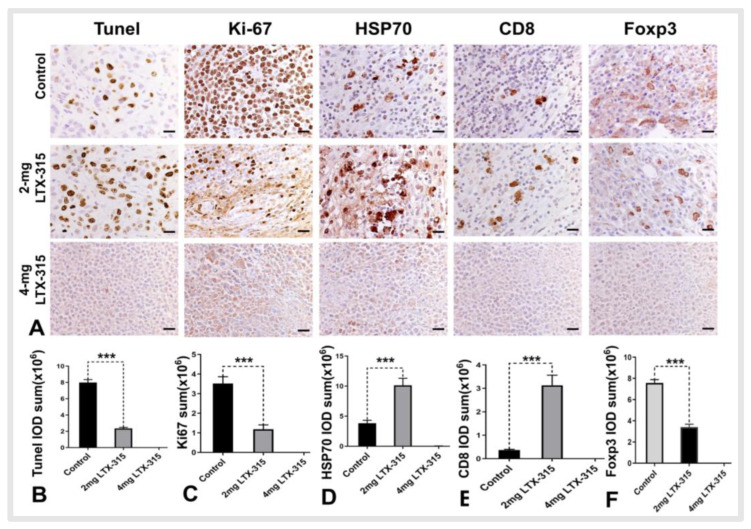
The immunohistochemical analysis of tumors among three treatment groups. (**A**) Due to the complete necrosis of tumor, there were no apoptotic cells, proliferative cells, HSP 70, CD8^+^ T cells, and Tregs in the ablated tumor areas in the 4 mg LTX-315 therapy group. (**A**–**C**) There were more apoptotic cells (yellow brown dots) and fewer proliferating cells (yellow brown dots) in the 2 mg LTX-315 therapy group than the control group. (**A**,**D**–**F**) The staining of HSP70, CD8, and Foxp3 demonstrated there were more HSP70 and CD8^+^ T cells and fewer Tregs in the residual tumors in the 2 mg LTX-315 therapy group than the control group. (Scale bars, 20 µm) (*** *p* < 0.001).

## Data Availability

All datasets of this study are available from the corresponding author on reasonable request.

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
