# Peer review of "Interventional Oncolytic Immunotherapy with LTX-315 for Residual Tumor after Incomplete Radiofrequency Ablation of Liver Cancer"

_cancers, 2022, doi:10.3390/cancers14246093_

Round 1

Reviewer 1 Report

Interesting and timely topic using an interesting molecule. In general well designed preclinical study.

Introduction:

Pls. provide more information on the problem of incomplete ablation (e.g. frequency, dependency on tumor location and ablation technology ...). Pls. clarify why you opted for combined treatement insted of a sequential therapy. Other options to treat incomplete ablation - e.g. TACE+RFA etc. should be mentioned.

Pls. provide more background information on LTX-315 and the current status on research on this particular drug. The term "first-in-class" should be avoided.

Methods:

Pls. provide the interval between VX2 implantation and treatment (average/range). 

Results:

Page 8 / Line240 - pls. calrify which tumor volume belongs to which treatment. The wording in the manuscript is equivocal.

Discussion:

The authors should discuss potential limits of drug distribution after percutaneous injection (vs. transarterial administration). Why did you opt for a local application instead of a systemic drug administration? The latter would be easier and more independent from the RF system (important as most ablation systems do not have dedicated injection channels)

Pls. discuss to what extent findings from a VX2 tumor model can be trensfered to clinical medicine - name potential limitations.

Author Response

Response to comments

Reviewer #1 (Reviewer Comments to the Author):

Introduction:

Pls. provide more information on the problem of incomplete ablation (e.g. frequency, dependency on tumor location and ablation technology ...). Pls. clarify why you opted for combined treatment instead of a sequential therapy. Other options to treat incomplete ablation - e.g. TACE+RFA etc. should be mentioned.

Response: We highly appreciate the reviewer’s comments. We have added more information about incomplete ablation in the revised manuscript. Unlike subcutaneous tumors, which can be treated by multiple injections, this experiment of rabbit liver cancer required laparotomy. In order to reduce trauma and animal pain, our institutional ethics policy  does not allow multiple invasive procedures on the same animal . Thus, we had to choose the combination therapy rather than sequential therapies. We have added the statement on other options to potentially treat incomplete ablation in the revised manuscript.

Pls. provide more background information on LTX-315 and the current status on research on this particular drug. The term "first-in-class" should be avoided.

  1. Response: According to the reviewer’s advice, we have added more background information on LTX-315 and the current status on scientific advances of it, as deleted the term "first-in-class."

Methods:

Pls. provide the interval between VX2 implantation and treatment (average/range).

Response:  The interval between VX2 implantation and treatment was two weeks, which has been clarified in Methods of the revised manuscript.

Results:

Page 8 / Line240 - pls. clarify which tumor volume belongs to which treatment. The wording in the manuscript is equivocal.

Response: We appreciate the reviewer’s comment, and have clarified that the tumor volume belongs to the treatment in the revised manuscript.

Discussion:

The authors should discuss potential limits of drug distribution after percutaneous injection (vs. transarterial administration). Why did you opt for a local application instead of a systemic drug administration? The latter would be easier and more independent from the RF system (important as most ablation systems do not have dedicated injection channels)

Response:  According to the reviewer’s comment, we have discussed potential limits of drug distribution after percutaneous injection (vs. transarterial administration). Since LTX-315  is rapidly hydrolyzed by peptidase after entering the blood, it is not ideal for systemic medication. Most critically, via systemic administration, therapeutics circulate through the entire body, with limited dose being delivered at the targets. With the RF ablation system, we can directly and locally deliver the highly concentrated dose of therapeutics into target, especially the difficult-to-treat ablated tumor margin, to facilitate the complete eradication of the residual tumors there. The technique we used in the current study presents the primary advantage of image-guided interventional treatment of the middle-to-large liver tumors. .

Pls. discuss to what extent findings from a VX2 tumor model can be transferred to clinical medicine - name potential limitations.

Response: Thank you for your suggestion. We discussed to what extent findings from a VX2 tumor model can be transferred to clinical medicine- name potential limitations  in the last paragraph of the discussion in the revised manuscript.

Reviewer 2 Report

This study investigated the feasibility of interventional oncolytic immunotherapy with LTX-315 for residual tumors after incomplete radiofrequency ablation (iRFA) of VX2 liver tumors in a rabbit model. The authors have provided sufficient evidences to support their conclusion on the subjects of concern. The work is interesting and may open new avenues to prevent residual tumors after RFA of intermediate-to-large liver cancers.

Abstract

1)      Please provide the full name of RFH when it first appeared.

2)       The parameters of RFH used in vitro study and the parameters of iRFA used in vivo study should be given.

Materials and methods :

1) Study design: All in-vitro experiments were repeated six times. What about in-vivo experiments?

2) The detail information of RFH treatment should be given, such as the frequency, the power density and so on. 

3) For RFH combined with LTX-315 group, were cells treated by RFH and LTX-315 at the same time? Please make it clear.

4) The detail information of VX-2 cell line should be given.

5)  Why female New Zealand rabbits were selected for this study? What about male rabbits?

Figures

1)      In Figure 6, which indicator can reflect the Tregs level?  Was it FOX-3? Please make it clear.

2)      All the morphological figures lack the explanation for scale bar.

Author Response

Response to comments

Reviewer #2 (Reviewer Comments to the Author):
Comments and Suggestions for Authors

This study investigated the feasibility of interventional oncolytic immunotherapy with LTX-315 for residual tumors after incomplete radiofrequency ablation (iRFA) of VX2 liver tumors in a rabbit model. The authors have provided sufficient evidences to support their conclusion on the subjects of concern. The work is interesting and may open new avenues to prevent residual tumors after RFA of intermediate-to-large liver cancers.

Abstract

1)      Please provide the full name of RFH when it first appeared.

Response:  We highly appreciate the reviewer’s comment, and have clarified the full name of RFH when it first appeared in the revised manusript.

2)       The parameters of RFH used in vitro study and the parameters of iRFA used in vivo study should be given.

Response: We have provided the parameters of RFH/iRFA used both in-vitro and in-vivo experiments in the revised manuscript.

Materials and methods:

1) Study design: All in-vitro experiments were repeated six times. What about in-vivo experiments?

Response:   For the in-vivo experiments, eighteen rabbits with residual VX2 tumors were randomly divided into 3 groups (n=6/group), which has been clarified in the revised manuscript.

2) The detail information of RFH treatment should be given, such as the frequency, the power density and so on.

Response:   We appreciate the reviewer’s comment, according to which we have added the detail information about the RFH treatment.

3) For RFH combined with LTX-315 group, were cells treated by RFH and LTX-315 at the same time? Please make it clear.

Response:  For RFH combined with LTX-315 group, LTX-315 was added to the well of tumor cells immediately followed by 30 min RFH at 42ºC, which has been clarified in the revised manuscript.

4) The detail information of VX-2 cell line should be given.

Response:  We have provided the detail information of VX-2 cell line.

5)  Why female New Zealand rabbits were selected for this study? What about male rabbits?

Response:  Since our experiment required two times of laparotomy. Considering that female rabbits are more gentle, it may reduce the risk of wound dehiscence or infection after surgery. Thus, we chose female rabbits for our study.

Figures

1)      In Figure 6, which indicator can reflect the Tregs level?  Was it FOX-3? Please make it clear.

Response:  In Figure 6, the Foxp3 can reflect the Tregs level. We have added more information about it in the revised manuscript.

2)      All the morphological figures lack the explanation for scale bar.

Response:   We highly appreciate the reviewer’s comment, and have explained the scale bar for all the morphological figures.
